Genomic relatedness and dissemination of blaNDM-5 among Acinetobacter baumannii isolated from hospital environments and clinical specimens in Thailand

Kitti Thawatchai 1
Manrueang Suphattra 2
Leungtongkam Udomluk 2
Khongfak Supat 2
http://orcid.org/0000-0003-2683-4284 Thummeepak Rapee 2
Wannalerdsakun Surat 3
Jindayok Thanyasiri 4
http://orcid.org/0000-0003-4992-7635 Sitthisak Sutthirat 2 5 sutthirats@nu.ac.th
1 Department of Oriental Medicine, ChiangRai College , Muang, Chiangrai , Thailand
2 Microbiology and Parasitology, Naresuan University , Muang, Phitsanulok , Thailand
3 Department of Internal Medicine, Faculty of Medicine, Naresuan University , Muang, Phitsanulok , Thailand
4 Department of Pathology, Faculty of Medicine, Naresuan University , Muang, Phitsanulok , Thailand
5 Centre of Excellence in Medical Biotechnology, Faculty of Medical Science, Naresuan University , Muang, Phitsanulok , Thailand
Franco Bernardo
Electronic publication date: 2023 Feb 7
Publication date: 2023
Volume: 11
Electronic Location ID: e14831
Received 2022 Oct 6; Accepted 2023 Jan 9
Copyright: © 2023 Kitti et al.
Copyright year: 2023
Copyright holder: Kitti et al.
License: This is an open access article distributed under the terms of the Creative Commons Attribution License, which permits unrestricted use, distribution, reproduction and adaptation in any medium and for any purpose provided that it is properly attributed. For attribution, the original author(s), title, publication source (PeerJ) and either DOI or URL of the article must be cited.
License URL: https://creativecommons.org/licenses/by/4.0/

Keywords: Acinetobacter baumannii, Hospital environment, Antibiotic resistance genes, blaNDM, Antibiotic resistance, Comparative genomes, Whole genome sequencing

Funding: National Science, Research and Innovation Fund R2564B016 The Royal Golden Jubilee Ph.D. Program PHD/0227/2560 This work was financially supported by National Science, Research and Innovation Fund (NSRF) 2020 (R2564B016). Udomluk Leungtongkam was supported by The Royal Golden Jubilee Ph.D. Program (PHD/0227/2560). The funders had no role in study design, data collection and analysis, decision to publish, or preparation of the manuscript.

==============================
Background

Acinetobacter baumannii (A. baumannii) is an important cause of nosocomial infection, especially in intensive care units (ICUs). It has the propensity to tolerate various environments and multiple classes of antibiotics. Our study aimed to characterize the comparative genomes of A. baumannii from hospital environments and clinical isolates.

Methods

Clinical and environmental A. baumannii isolates were collected from a university hospital. Antibiotic susceptibility testing was performed, antibiotic resistance genes (ARGs) were characterized, and repetitive element palindromic-PCR (rep-PCR) typing was performed. Eight representative A. baumannii isolated from environmental and clinical samples from the same wards were selected for whole-genome sequencing (WGS) using the Illumina platform.

Results

A total of 106 A. baumannii isolates were obtained from 312 hospital environmental samples. A high percentage of samples with A. baumannii colonization were detected from AMBU bags (77.9%), followed by bedrails (66.7%) and suction tubes (66.7%). We found that 93.4% of the environmental isolates were multidrug-resistant A. baumannii (MDRAB), and 44.7% were extremely drug-resistant A. baumannii (XDRAB). blaOXA-23 blaNDM, and blaOXA-58 were present in 80.2%, 78.3%, and 0.9% of all isolates, respectively. Sixty-one A. baumannii isolates were collected from patient specimens in the same ward. Among all A. baumannii clinical isolates, MDRAB and XDRAB accounted for 82% and 55.7%, respectively. The most dominant ARGs identified was blaOXA-23 (80.3%), followed by blaNDM (55.7%). The genetic diversity of all isolates using rep-PCR could be divided into 33 genotypes. The genome size of eight A. baumannii ranged from 3.78–4.01 Mb. We found six of eight strains to be blaNDM-5-harboring A. baumannii. Mobile genetic elements (MGEs), such as integron1 (intl1), located upstream of blaNDM-5 were observed. The phylogenomic relationship of the core and pan genomes as well as the single nucleotide polymorphism (SNP) count matrix revealed the genetic similarity of A. baumannii environmental and clinical strains obtained from the same ward.

Conclusion

This study confirmed that A. baumannii colonized in hospital environments were the main reservoir of nosocomial infection and provides critical information to guide the control of A. baumannii infection.

Introduction

Acinetobacter baumannii has emerged as an important pathogen related to hospital-acquired infections worldwide. This pathogen is the major cause of ventilator-associated pneumonia (VAP), bacteremia, urinary tract infections, wound infections, and meningitis (Nutman et al., 2016). The emergence of antibiotic-resistant A. baumannii, especially MDRAB and XDRAB, has increased and seriously challenged the treatment of these bacterial infections (Kyriakidis et al., 2021). National Antimicrobial Resistance Surveillance Thailand (NARST) reported that the prevalence of carbapenem-resistant Acinetobacter baumannii complex infection in the ICUs of 51 hospitals in Thailand was higher than 80% (NARST, 2021). The major mechanism of carbapenem resistance among A. baumannii is the production of antibiotic-hydrolyzing enzymes that belong to Ambler Class D β-lactamases (CHDLs) and class B metallo-lactamases (MBLs) (Ibrahim et al., 2021). Class D carbapenemases encode acquired blaOXA-23, blaOXA-24, and blaOXA-58. These genes have been reported in many countries all over Asia, including China, Korea, Thailand, Vietnam, and Malaysia (Hsu et al., 2017). Major MBLs in A. baumannii are encoded by the blaNDM gene and has been reported in Thailand since 2017 (Leungtongkam et al., 2018). To date, twenty-four New Delhi metallo-beta-lactamase (NDM) variants have been identified in more than 60 bacterial species, including Acinetobacter spp., and several variants have the ability to enhance carbapenemase activity (Wu et al., 2019).

A. baumannii has the ability to survive on hospital surfaces and equipments for long periods. Hospital surface contamination of A. baumannii is closely correlated with the transmission of the bacteria to patients, causing episodes of bacteremia and/or sepsis (Markogiannakis et al., 2008). Genome sequencing of carbapenem-resistant A. baumannii (CRAB) found on ICU surfaces revealed that the CRAB isolates from ICU environment were linked with those of clinical origin (Yasir et al., 2022). A. baumannii isolates were recovered from surrounding ICU bed surfaces, and these isolates exhibited a multidrug resistance phenotype and belonged to some widely spread clonal complexes (CCs) of clinical A. baumannii isolates (Rocha et al., 2018).

Comparative genomics research can help assess the bacterial evolution, resistance mechanisms, and pathogenicity of bacterial pathogens at the genome-wide level; it is also useful in the ensuing study of virulence factors involved in pathogenicity (Wright et al., 2016). Whole-genome sequencing studies comparing distinct clinical and environmental isolates have improved our understanding of the evolution of A. baumannii. In this study, we aimed to investigate the resistance rates and epidemiological characteristics of clinical and environmental A. baumannii isolates. Then, we determined the draft genome sequence of eight clinical and eight environmental A. baumannii strains from the same wards to perform comparative genomic analysis.

Materials and Methods

Samples

Clinical and environmental A. baumannii isolates were collected from Naresuan University Hospital between December 2020 and April 2021. Naresuan University is a level III hospital with 400 beds located in the lower northern region of Thailand. Hospital environment and clinical isolates were collected from five wards, which were two medical wards, Medicine-man (MED-1) and Medicine-woman (MED-2), and three intensive care units, the ICU Cardio-Vascular-Thoracic Surgery (ICU-1), ICU Surgery (ICU-2), and ICU Medicine (ICU-MED). The sources of the samples included staff contact samples, which included samples collected from stethoscopes (n = 15), charts (n = 15), computers/keyboards (n = 15), nurse station counters (n = 15), medical lab coats (n = 15), restroom door handles (n = 15), telephones (n = 15), and dressing trolleys (n = 15). Patient contact samples were collected from bedrails (n = 15), bedsheets (n = 15), suction tubes (n = 15), patient tables (n = 15), curtains (n = 15), humidifiers (n = 15), intravenous (IV) stands (n = 15), ventilators (n = 15), ventilator monitors (n = 9), water from ventilators (n = 9), suction tubes (n = 9), and AMBU bags (n = 9). Other environmental samples were collected from the air (n = 15), sinks (n = 15), and water from sinks (n = 15). The protocol was approved by the Naresuan University Institutional Biosafety Committee, and the project number was NUIBC MI62-09-42.

Isolation and identification of A. baumannii from hospital environments

The air samples were collected using Leeds Acinetobacter Medium (LAM) (Hi-media, Mumbai, Maharashtra, India) in 9 cm diameter Petri dishes. Petri dishes were exposed for 24 h. The other samples from environmental surfaces were collected using cotton swabs soaked in 0.85% normal saline and then placed in transfer media. The swab samples were enriched in Luria-Bertani broth (LB) (HiMedia, Mumbai, Maharashtra, India) by shaking at 160 rpm at 37 °C for 24 h and then cultured in Leed Acinetobacter Media (LAM) at 37 °C for 24 h. Cultures with pink colonies were selected for further evaluation using Gram’s stain and biochemical tests (catalase, oxidase, TSI, citrate). Molecular identification of the bacterial isolates was confirmed by polymerase chain reaction (PCR) using 16S rRNA, and blaOXA-51 primers (Table S1).

Determination of antibiotic susceptibility

Antibiotic susceptibility testing was performed according to the disk diffusion method using 12 antibiotics: piperacillin/tazobactam (100 and 10 μg), ceftazidime (30 μg), cefepime (30 μg), cefotaxime (30 μg), ceftriaxone (30 μg), imipenem (10 μg), meropenem (10 μg), gentamicin (10 μg), amikacin (30 μg), tetracycline (30 μg), ciprofloxacin (5 μg), and trimethoprim/sulfamethoxazole (1.25 and 23.75 μg). The plates were incubated at 37 °C for 24 h. The zones of inhibition determined whether the microorganism was susceptible, intermediate, or resistant to each antibiotic according to the Clinical and Laboratory Standards Institute (CLSI) guidelines (2022). All isolates were interpreted as non-drug-resistant A. baumannii (NRAB) and carbapenem-resistant A. baumannii (CRAB). In addition, MDRAB was defined when A. baumannii resistant to three or more antibiotic classes and XDRAB was defined when A. baumannii was resistant to all antimicrobial agents except polymyxins (Magiorakos et al., 2012).

PCR amplification of antibiotic resistance genes and rep-PCR typing

As mentioned earlier, PCR assays to detect blaOXA-23, blaOXA-24, blaOXA-58, and blaNDM were performed using the primers shown in Table S1. The genomic DNA of each isolate was extracted from the overnight cultures using a PureDirex Genomics DNA Isolation Kit (Bio-Helix, New Taipei City, Taiwan). Rep-PCR was performed by using genomic DNA as a template for PCR amplification with the ERIC-2 primer (Table S1) with the conditions described by Leungtongkam et al. (2018). PCR-banding patterns and rep-PCR types were analyzed and interpreted as previously described.

Whole-genome sequencing and bioinformatics analysis

Eight representative A. baumannii strains from four wards, four from hospital environments (AE17, AE30, AE73, AE106) and four from clinical isolates (AC02, AC09, AC23, and AC59) were analyzed. We selected two A. baumannii strains from each ward that were isolated from the same time frame and showed similar antibiotic susceptibility profiles and ARG patterns. All strains were cultured onto an LB agar plate and incubated overnight at 37 °C. Genomic DNA was extracted using a PureDire Genomics DNA Isolation Kit (Bio-Helix, New Taipei City, Taiwan). The extracted DNA was quantified by a nanodrop (Hercuvan, Cambridge, UK). The purified genomic DNA was used to construct libraries followed by sequencing with the Illumina HiSeq 2500-PE125 platform at Macrogen, Korea. The nucleotide sequences of the eight A. baumannii strains have been deposited in NCBI’s database under Sequence Read Archive (SRA) with Bioproject PRJNA862456. The genome of A. baumannii ATCC17978 (CP000521) was used as a reference strain for comparison with the eight A. baumannii strains.

Genome assembly and annotation

Raw sequencing reads were trimmed by using Trim Galore v0.6.7 with default settings and by using Unicycler v0.4.8 with default parameters prior to assembly (Krueger, 2012; Wick et al., 2017). The assembled contigs that were larger than 300 bp in length were selected and subjected to further bioinformatic analysis. The remaining contigs were annotated by using Prokka v1.14.6 with default options (Seemann, 2014).

Identification of MLST, antimicrobial resistance, and virulence genes

The remaining contigs were subjected to detection of drug-resistance and virulence genes by using Abricate v1.0.1 with default settings (Seemann, 2016) against the comprehensive antibiotic resistance database (CARD) and virulence factor database (VFDB) (Alcock et al., 2020; Liu et al., 2022). Multilocus sequence typing (MLST) were performed by using MLST v2.0, which is accessible from the Center for Genomic Epidemiology (www.genomicepidemiology.org). The gene arrangement analysis of blaNDM-5 was performed using Easyfig version 2.1 (Sullivan, Petty & Beatson, 2011).

Phylogenomic relationships

The selected genomes of eight A. baumannii were subjected to Roary v3.13.0 with the default parameters to identify pan- and core genes (Page et al., 2015). The resultant core genes among the eight genomes were concatenated prior to the construction of a pangenome tree in the CSI phylogeny, which is accessible from the Center for Genomic Epidemiology (www.genomicepidemiology.org) (Kaas et al., 2014). A core-genome tree was constructed based on the presence/absence of identified core-genes and visualized in FigTree v1.4.4 (http://tree.bio.ed.ac.uk/software/figtree/). The SNP count matrix of all selected genomes was calculated in snp-dists v0.6.3 with default settings (Seemann, 2019).

Statistical analysis

Statistical analysis were performed using Stata (Stata 12.0 Corporation). The comparisons of the proportions of antibiotic resistance between A. baumannii obtained from the two different origins were analyzed by using the Z-test. The comparisons of antibiotic resistance among A. baumannii collected from the five hospital wards were analyzed by using the chi-square test. P values < 0.05 were considered to be a statistically significant difference.

Results

A. baumannii strains isolated from the hospital environments and clinical isolates

A total of 106 A. baumannii isolates were obtained from 312 hospital environmental samples (33.97%). The isolates associated with patient contact from AMBU bags, bedrails, suction tubes, water from ventilators, bedsheets, and IV stands were found in 77.9%, 66.7%, 66.7%, 55.6%, 53.3%, and 13.3% of the samples, respectively. We also found 33.3% of the samples from patient tables, humidifiers, ventilators, and curtains were A. baumannii colonization (Table S2). The isolates associated with staff contact and other environments from the air, keyboards, counters, medical lab coats, and dressing trolleys were found in 60.0%, 53.3%, 46.7%, 42.9%, and 33.3% of the samples, respectively (Table S2). The colonization rate of the samples from charts and stethoscopes was 26.7%, while 6.7% of the samples from restroom door handles, and telephones were A. baumannii colonization. However, we did not find A. baumannii isolates on sinks, water from sinks, or ventilator monitors (Table S2). Of the 312 environmental samples collected from each ward, we found the highest A. baumannii contamination in the samples obtained from ICU Surgery, with a rate of 52.9% (36/38), followed by those obtained from the Medicine-woman (40.7%; 22/54), ICU Medicine (38.2%; 26/68), Medicine-man (27.8%; 5/54), and ICU Cardiovascular-Thoracic Surgery (10.3%; 7/68) wards (Table S2).

During the investigation of the prevalence of A. baumannii isolates from the hospital environments of various wards, we found the highest rate of A. baumannii in the ICU Surgery ward (33.9%), followed by the ICU Medicine (24.5%), Medicine-woman (20.8%), Medicine-man (14.2%), and ICU Cardio-Vascular-Thoracic surgery (6.6%) wards (Table 1). Sixty-one A. baumannii isolates were collected from patient specimens. A. baumannii isolates were found in the patient specimens collected from the ICU Medicine (24.6%), Medicine-man (24.6%), ICU Surgery (19.7%), Medicine-woman (16.4%), and ICU Cardio-Vascular-Thoracic surgery (14.8%) wards (Table 1).

Table 1 A. baumannii isolated from hospital environments and clinical samples from various hospital wards.

Ward	Positive environment	Positive clinical	
n	%	n	%	
MED-1	Medicine-man ward	15	14.2%	15	24.6%	
MED-2	Medicine-woman ward	22	20.8%	10	16.4%	
ICU-MED	ICU medicine	26	24.5%	15	24.6%	
ICU-1	ICU cardio-vascular-thoracic surgery	7	6.6%	9	14.8%	
ICU-2	ICU surgery	36	33.9%	12	19.6%	
Total	106	100.00%	61	100.00%	

Antibiotic susceptibility patterns of A. baumannii isolates

All A. baumannii isolates were subjected to antimicrobial susceptibility testing, and the results are shown in Table 2. A. baumannii isolates from hospital environments were highly resistant to meropenem (100%), cefotaxime (100%), ceftazidime (100%), and ceftriaxone (100%), while the A. baumannii clinical isolates were highly resistant to ceftazidime (90.2%) and ceftriaxone (90.2%). NRAB was detected in only 16.39% of A. baumannii clinical isolates. A high prevalence of MDRAB and CRAB was detected in A. baumannii isolated from hospital environment (ABHE) (93.4% and 100%) and clinical isolates (82.0% and 92.0%) with p value < 0.05, as shown in Table 3. The prevalence of XDRAB in A. baumannii isolates from hospital environments and clinical isolates was 44.7% and 55.7%, respectively (Table 3). Among the five wards, a high prevalence of XDRAB was detected in A. baumannii isolates from ICU Surgery (Table 4).

Table 2 Frequency of resistance to antimicrobial agents among A. baumannii isolates from hospital environments and clinical samples.

Antimicrobial group	Antibiotics	Resistance		
		Hospital
environment	Clinical	
β-Lactam combinations	Piperacillin/Tazobactam	80.2%	81.9%	
Cephems	Ceftazidime	100.0%	90.2%	
	Cefepime	99.1%	85.3%	
	Cefotaxime	100.0%	88.3%	
	Ceftriaxone	100.0%	90.2%	
Carbapenems	Imipenem	77.4%	55.7%	
	Meropenem	100.0%	83.6%	
Aminoglycosides	Gentamicin	77.4%	70.5%	
	Amikacin	62.3%	67.2%	
Tetracyclines	Tetracycline	74.5%	73.8%	
Fluoroquinolones	Ciprofloxacin	79.2%	83.6%	
Folate pathway inhibitors	Trimethoprim/Sulfamethoxazole	88.7%	81.9%	

Table 3 The statistical analysis for comparing the proportions of antibiotic resistance between A. baumannii obtained from two different origins.

Characteristics	Clinical origin
(n = 61 isolates)	Environmental origin
(n = 106 isolates)	*p value (95% CI)	
Prevalence of MDRAB	50/61 (82.0%)	99/106 (93.4%)	0.021 [22.2–0.7%]	
Prevalence of CRAB	50/61 (92.0%)	106/106 (100%)	<0.001 [83.8–27.7%]	
Prevalence of XDRAB	34/61 (55.7%)	47/106 (44.7%)	0.116 [27.0–4.2%]	
Prevalence of blaOXA-23 positive isolates	49/61 (80.3%)	85/106 (80.2%)	0.983 [12.4 to –12.7%]	
Prevalence of blaOXA-58 positive isolates	1/61 (1.6%)	1/106 (0.9%)	**ND	
Prevalence of blaNDM positive isolates	34/61 (55.7%)	83/106 (78.3%)	0.002 (37.3–7.8%)	
Notes:

* Comparison of percentages between two groups by Z-test.

** ND, Not determined statistical analysis.

A p value < 0.05 reflected statistically significant findings. CRAB, carbapenem-resistant A. baumannii; MDRAB, multidrug-resistant A. baumannii; XDRAB, extremely drug-resistant A. baumannii.

Table 4 Proportion comparisons of antibiotic resistance among A. baumannii collected from five hospital wards.

Hospital wards/characteristics	MED-1	MED-2	ICU-MED	ICU-1	ICU-2	*p value	
Percentage of MDRAB	29/30 (96.7%)	26/32 (81.3%)	37/41 (90.2%)	15/16 (93.3%)	42/48 (87.5%)	0.386	
Percentage of CRAB	29/30 (96.7%)	31/32 (96.9%)	39/41 (95.1%)	15/16 (93.8%)	42/48 (87.5%)	0.490	
Percentage of XDRAB	13/30 (43.3%)	11/32 (34.4%)	13/41 (31.7%)	8/16 (50%)	36/48 (75%)	<0.001	
Percentage of blaOXA-23 positive isolates	27/30 (90%)	14/32 (43.8%)	36/41 (87.8%)	15/16 (93.8%)	42/48 (87.5%)	<0.001	
Percentage of blaOXA-58 positive isolates	1/30 (3.3%)	0/32 (0%)	0/41 (0%)	0/16 (0%)	1/48 (2.1%)	**ND	
Percentage of blaNDM positive isolates	14/30 (46.7%)	29/32 (90.6%)	22/41 (53.7%)	8/16 (50%)	44/48 (91.7%)	<0.001	
Note:

* Overall p value calculated to compare percentages among multiple groups by Chi-square test.

** ND, Not determined statistical analysis.

Bold values denote the highest proportions with statistical significance at the p value < 0.05 level. MED-1, Medicine-man ward; MED-2, Medicine-woman ward; ICU-MED, ICU Medicine; ICU-1, ICU Cardio-Vascular-Thoracic Surgery; ICU-2, ICU Surgery; CRAB, carbapenem-resistant A. baumannii; MDRAB, multidrug-resistant A. baumannii; XDRAB, extremely drug-resistant A. baumannii.

Antibiotic resistance genes and rep-PCR typing

The 16S rRNA gene was detected in all A. baumannii isolates. The intrinsic blaOXA-51 gene was detected in all ABHE and 96.7% (59/61) of clinical isolates. The oxacillinase gene, blaOXA-23 was the most frequently detected gene at 80.20% (85/106) in ABHE and 80.33% (49/61) in clinical isolates (Table 3). The blaOXA-58 gene was detected in one ABHE (0.94%) and one clinical isolate (1.64%). The blaNDM gene was detected in 78.3% (83/106) of ABHE (p value < 0.05) compared to 55.74% (34/61) of clinical isolates. The blaOXA-24 gene was not detected in any of the isolates. Among the five wards, a high prevalence of blaOXA-23 was detected in ICU Cardio-Vascular-Thoracic Surgery, and a high prevalence of blaNDM was detected in ICU Surgery (p value < 0.05) (Table 4).

Rep-PCR typing was performed, and fingerprinting represented 33 different DNA patterns consisting of amplicon sizes ranging from 500 to 4,000 bp. The genotypes were named T1 to T33. The major genotype of ABHE was T30 at 21.7% (23/106), followed by T23 at 17% (18/106) and T2 at 15% (15/106). The major genotype of the A. baumannii clinical isolates was T4 at 34.4% (21/61), followed by T23 at 29.5% (18/61).

Heatmaps representing the antibiotic susceptibility patterns, antimicrobial resistance genes, and rep-PCR typing from the five wards is shown in Figs. S1–S5. We found genetic similarity between ABHE and A. baumannii clinical isolates in each ward with antibiotic susceptibility patterns and antimicrobial resistance genes since most A. baumannii strains in the same ward showed similar profiles. No association was found between rep-PCR typing of ABHE and A. baumannii clinical isolates (Figs. S1–S5). Eight strains of A. baumannii with similar profiles from four wards were selected for genome sequencing.

Comparative genomic and phylogenomic analysis of A. baumannii from hospital environmental and clinical isolates

Eight strains of A. baumannii from clinical and environmental isolates were analyzed and compared with the genome of A. baumannii ATCC17978. The four ABHE were AE17 (patient table), AE30 (bedrail), AE73 (dressing trolley), and AE106 (AMBU bag). The four clinical isolates were AC02 (blood hemoculture), AC09 (sputum), AC23 (sputum), and AC59 (right hepatic drain). AC02 and AE03 were obtained from the Medicine-man ward. AC59 and AE17 were obtained from the Medicine-woman ward. AC09 and AE106 were derived from the ICU Cardio-Vascular-Thoracic Surgery ward. AC23 and AE73 were derived from the ICU Surgery ward. The genome characterization of the isolates is summarized in Table 5. The genome analysis revealed that AC02, AE30, AC09, AE106, AC23 and AE73 belong to ST2 based on the Pasteur MLST scheme. However, AC59 and AE17 belong to ST164. The predicted genome sizes of the eight A. baumannii strains ranged from 3.78 to 4.01 Mb compared to the genome of ATCC17978, which was 3.97 Mb.

Table 5 Medical and general genome features of eight representatives A. baumannii isolated from various hospital wards.

Strain ID/
characteristics	AC02	AE30	AC59	AE17	AC09	AE106	AC23	AE73	
Ward	MED-1	MED-1	MED-2	MED-2	ICU-1	ICU-1	ICU-2	ICU-2	
Specimen types	Blood-hemoculture	Bedrail	Sputum	Patient table	Sputum	AMBU bag	Right hepatic drain	Dressing trolley	
Antibiotic resistance	XDRAB	XDRAB	MDRAB	MDRAB	XDRAB	XDRAB	MDRAB	MDRAB	
MLST	ST2	ST2	ST164	ST164	ST2	ST2	ST2	ST2	
Genome size (bp)	4,016,797	3,966,329	3,958,580	3,786,785	3,934,990	3,949,273	3,925,340	3,955,274	
% GC	38.90	38.99	38.87	38.88	38.98	39.00	38.98	38.99	
No. of contigs	86	71	96	63	68	76	72	81	
Largest contig	340,426	292,477	481,102	306,399	303,352	292,477	360,663	292,477	
Note:

MED-1, Medicine-man ward; MED-2, Medicine-woman ward; ICU-1, ICU Cardio-Vascular-Thoracic Surgery; ICU-2, ICU Surgery; MDRAB, multidrug-resistant A. baumannii; XDRAB, extremely drug-resistant A. baumannii.

ARGs and virulence genes of eight A. baumannii strains showed genetic similarity among A. baumannii hospital environments and clinical isolates but were slightly different from the genome of ATCC17978 (Figs. 1A and 1B). The ARGs detected in all eight A. baumannii strains as well as ATCC 17978 encoded macrolide resistance genes (amvA) and a number of genes encoding efflux pumps involved in resistance in glycylcycline/tetracycline (adeR, adeS, adeA, adeB), fluoroquinolone/tetracycline (adeF, adeG, adeH, adeL), fluoroquinolone (abaQ, abeM), fosfomycin (abaF), and multidrug resistance (adeI, adeJ, adeK, adeN, abeS). We identified 23 ARGs present in only some A. baumannii strains, which encoded the efflux pump (adeC) and genes involved in resistance to tetracycline (tet(39), tetB), cephalosporins (blaADC-10, blaADC-6, blaADC-73, blaADC-79, blaTEM-1, blaTEM-12), carbapenems (blaOXA-23, blaOXA-66, blaOXA-91, blaOXA-259), macrolide (mphE, msrE), aminoglycoside (aadA5, armA, aph(3′′)-Ib, aph(6)-Id), sulfonamide (sul1, sul2), and integron-encoded dihydrofolate reductase (dfrA17).

Figure 1 Detections of antibiotic resistance, virulence genes, and genetic contexts of A. baumannii harboring blaNDM-5 among eight representative A. baumannii strains and ATCC 17978.

(A) The pattern of acquired resistance genes, (B) virulence factor-associated genes in the A. baumannii genomes, and (C) genetic contexts and comparison of the gene arrangement of six A. baumannii isolates harboring blaNDM-5. The arrows indicate genes located upstream and downstream of blaNDM-5, including Integron1 (intl1), BsuBI-PstI family restriction endonuclease (Bsu-PstI), aminoglycoside 3″-nucleotidyltransferase (ant(3″)-Ia), quaternary ammonium compound efflux (qacEΔ1), sulfonamide resistance (sul1), IS91 family transposase, cytochrome c-type biogenesis protein (DsbD), N-(5′-phosphoribosyl) anthranilate isomerase (trpF), bleomycin resistance protein (bleMBL), New Delhi metallo-beta-lactamase 5 (blaNDM-5), and transposase (ISAba125).

A class B β-lactamase gene, blaNDM-5, that hydrolyzes virtually all β-lactam antibiotics, including carbapenems, was detected in six strains except ATCC17978, AE17 and AC59 (Figs. 1A and 1B). Genetic contexts of blaNDM-5 revealed mobile genetic elements (MGEs), such as integron1 (intl1), IS91 family transposase, and transposase (ISAba125), along with other AGRs, ant(3″)-Ia, qacEΔ1, and sul1, located upstream and downstream of blaNDM-5 (Fig. 1C).

Analysis of the virulence genes of eight A. baumannii strains and ATCC17978 revealed that the genes were involved in biofilm formation (adeF, adeG, adeH, bap, csuA/B, csuA, csuB, csuC, csuD, csuE, pgaA, pgaB, pgaC, pgaD), enzyme phospholipase (plcC, plcD), immune evasion (lpsB, lpxA, lpxB, lpxD, lpxL, lpxM), iron uptake (barA, barB, basA, basB, basC, basD, basF, basG, basI, basJ, bauA, bauB, bauC, bauD, bauE, bauF, entE), gene regulation (abal, abaR, bfmR, bfmS), serum resistance (pbpG), and host cell adherence (ompA) (Fig. 1B). The genes involved in capsule polysaccharide synthesis (weoB) and the gene encoding glycosyltransferase in lipopolysaccharide (LPS) biosynthesis (lpsB) were detected in only one strain, ATCC 17978 and AC09 (Fig. 1B).

The phylogenomic relationship of the core and pan genomes of eight A. baumannii and ATCC17978 strains shown in Figs. 2A and 2B revealed three major clades. The A. baumannii strains obtained from the ICU-1, ICU-2, and Med-1 wards were in the same clade, while the A. baumannii strains obtained from the Med-2 ward were in different clades. The genome of ATCC17978 showed different clades from all eight A. baumannii strains. The SNP count matrix of all selected genomes confirmed that the high number of SNPs of AC59 and AE17 derived from the Med-2 ward were comparable with other A. baumannii strains (Fig. 2C).

Figure 2 Phylogenomic relationship among selected representative isolates of Acinetobacter baumannii obtained from different wards.

(A) A phylogeny reconstructed from 2,928 concatenated core genes of all analyzed genomes presented with metadata. (B) Hierarchical tree based on the presence/absence of patterns of 4,778 pangenome genes of eight representative isolates and ATCC 17978. (C) SNP matrix-based heatmap illustrating the number of single nucleotide polymorphisms in the whole genome between the eight strains studied.

Discussion

A. baumannii is an opportunistic pathogen that causes hospital-acquired infections in patients who have high risk factors, such as patients in intensive care units (ICUs). This bacterium is extremely capable of surviving, spreading, and developing antibiotic resistance in hospital wards (Vázquez-López et al., 2020). In this study, we investigated A. baumannii from three ICUs and two medicine wards from a university hospital to identify nosocomial infection-associated bacteria. A total of 106 isolates of A. baumannii were isolated from 312 environmental samples, which were frequently in contact with staff and patients. The highest numbers of staff and patient contact samples with A. baumannii colonization were from AMBU bags (77.9%) and keyboards (53.3%). Shamsizadeh et al. (2017) reported that A. baumannii was detected in environmental samples with the highest recovery in intensive care units (ICUs). This is in agreement with our study in which we isolated the highest number of A. baumannii from two ICUs. A previous study demonstrated that A. baumannii was isolated from hospital sinks, bed rails, water systems, and medical equipment, particularly in ICUs and surgical units (Ibrahim et al., 2021). We detected a high number of A. baumannii from AMBU bags (77.9%), followed by bedrails (66.7%) and suction tubes (66.7%). However, we did not obtain A. baumannii from hospital sinks or water from sinks. In addition, a previous study reported that the airborne route also plays an important role in the transmission of A. baumannii infections in hospitals (Ayoub Moubareck & Hammoudi Halat, 2020). Our study confirmed that a high number of A. baumannii was isolated from air (60.0%). A. baumannii was associated with hospital-acquired outbreaks due to its ability to spread in the air environment and colonize hospital utensils.

MDRAB and CRAB were described as major resistant strains that caused hospital outbreaks in Thailand (Leungtongkam et al., 2018; Chukamnerd et al., 2022).

High prevalence rates of both MDRAB and CRAB were found in this study. We found that the resistance rate of A. baumannii isolated from hospital environments was higher than that isolated from clinical samples. In addition, all A. baumannii isolates isolated from hospital environments were resistant to meropenem (100%), cefotaxime (100%), ceftazidime (100%), and ceftriaxone (100%), and all isolates were CRAB. The results were in contrast with a Chinese study showing that A. baumannii isolated from the hospital environment was more susceptible to most antimicrobial agents (Ying et al., 2015).

A. baumannii harboring blaOXA-51 gene has been identified as a marker for species identification. We detected blaOXA-51 gene in all environmental isolates but found two isolates from clinical specimens were blaOXA-51 gene negative. Further study is needed to identify different Acinetobacter species in these two strains. Our data showed that A. baumannii isolated from hospital environments and clinically isolated from the same ward possessed similar antibiotic susceptibility profiles, and ARG patterns represented the outbreak clone in each ward (Figs. S1–S5). Among all isolates, the results showed that blaOXA-23 was the most frequent carbapenemase gene detected. This result suggests that blaOXA-23 was the major cause of carbapenem resistance in A. baumannii isolates from hospital environments and clinical samples in our hospitals. This result was supported by Leungtongkam et al. (2018), who detected blaOXA-23 in all A. baumannii isolates from four tertiary hospitals in Thailand. Jain et al. (2019) reported that blaNDM-1 was the most frequent gene detected in A. baumannii isolated in both clinical and environmental samples from India (Jain et al., 2019). Interestingly, we found a high prevalence of blaNDM among both the hospital environment and clinical sample isolates. Compared to a previous report from Thailand, a low rate of blaNDM was detected in A. baumannii isolates from hospitals in northern and southern Thailand (Leungtongkam et al., 2018; Chukamnerd et al., 2022).

Genomic analysis of eight representative MDRAB strains found that the major ST type (AC02, AE30, AC09, AE106, AC23, and AE73) was ST2. It has been reported that MDRAB sequence type ST2 was the most prevalent in Thailand. The AC59 and AE17 strains were designated ST164, which was also reported in Thailand (Khuntayaporn et al., 2021). NDM-producing organisms have become endemic in the Indian subcontinent, and numerous epidemics have been recorded worldwide. Genomic analysis found that the AC02, AE30, AC09, AE106, AC23, and AE73 strains possess an NDM-5 metallo-β-lactamase gene. This is the first report regarding the detection of an NDM-5-producing A. baumannii from hospital environments and clinical samples in Thailand. The emergence of the blaNDM-5 gene was mostly identified in Escherichia coli. To date, only one report by (Khalid et al., 2020) identified A. baumannii harboring blaNDM-5 from the neonatal intensive care unit (NICU) of an Indian Hospital, but it was not present in environmental isolates (Khalid et al., 2020). Our PCR study identified the blaNDM gene but could not specifically identify the NDM variant. The outbreak clone harboring blaNDM-5 was revealed using WGS. Mobile genetic elements such as insertion sequences, transposons, and integrons can mobilize blaNDM-5 (Wu et al., 2019). Our WGS analysis revealed intl1 located upstream of blaNDM-5 (Fig. 1C). A previous report on E. coli detected blaNDM-5 to be located in a complex of class 1 integrons together with aadA2, aac(3)-IIa, mph(A), sul1, tet(A), and dfrA12 (Alba et al., 2021). In this study, we found ant(3″)-Ia, qacEΔ1, and sul1.

WGS of eight strains revealed a high number of ARGs in accordance with previous reports in Thailand (Kongthai et al., 2021; Wareth et al., 2021; Chukamnerd et al., 2022). Among the eight strains, the antibiotic resistance gene patterns of A. baumannii differed among wards but were similar in the same ward. A high number of acquired ARGs was detected. Horizontal gene transfer among A. baumannii and other bacterial species colonizing the hospital environment may play an important role in the movement of these acquired ARGs. Interestingly, we found that the virulence gene patterns of A. baumannii strains from four wards were quite similar (Fig. 1B). These findings indicated that all A. baumannii strains from the four wards were derived from the same ancestor and employed the same pathogenic mechanisms to cause disease. The phylogenomic relationship of the core and pan genomes as well as the SNP count matrix revealed the genetic similarity of A. baumannii strains obtained from the same ward. This is in agreement with a previous study by Yasir et al. (2022), in which genome sequencing revealed that A. baumannii isolated from hospital environments was linked with those of clinical origin (Yasir et al., 2022).

Conclusions

In conclusion, in this study, we presented a whole-genome analysis of eight A. baumannii strains from hospital environments and clinical samples. Our data revealed the epidemiological characteristics of similar antibiotic susceptibility profiles, antibiotic resistance genes, virulence genes, clonal complexes, core genomes, pan genomes, and single nucleotide polymorphisms among clinical and environmental A. baumannii isolates from the same ward.

Supplemental Information

Supplemental Information 1 List of primers used in this study.

Click here for additional data file.

Supplemental Information 2 A. baumannii strains isolated form hospital environment samples.

Click here for additional data file.

Supplemental Information 3 Resistance pattern and resistance genes from A. baumannii enviroment and clinical isolates.

Click here for additional data file.

Supplemental Information 4 Antibiotic resistance genes of eight A. baumannii isolates.

Click here for additional data file.

Supplemental Information 5 virulence genes of eight A. baumannii isolates.

Data from WGS identified virulence genes of eight A. baumannii isolates

Click here for additional data file.

Supplemental Information 6 Heatmap of antibiotic resistance patterns, antibiotic resistance genes, and rep-PCR typing of A. baumannii isolated from Medicine-woman ward (MW-MED2).

Click here for additional data file.

Supplemental Information 7 Heatmap of antibiotic resistance patterns, antibiotic resistance genes, and rep-PCR typing of A. baumannii isolated from Medicine-man ward (MM-MED1).

Click here for additional data file.

Supplemental Information 8 Heatmap of antibiotic resistance patterns, antibiotic resistance genes, and rep-PCR typing of A. baumannii isolated from ICU Medicine (IM-IMED).

Click here for additional data file.

Supplemental Information 9 Heatmap of antibiotic resistance patterns, antibiotic resistance genes, and rep-PCR typing of A. baumannii isolated from ICU Surgery (IS-ICU2).

Click here for additional data file.

Supplemental Information 10 Heatmap of antibiotic resistance patterns, antibiotic resistance genes, and rep-PCR typing of A. baumannii isolated from ICU Cardiovascular-Thoracic Surgery (IC-ICU1).

Click here for additional data file.

The authors would like to thank the staffs of Naresuan University hospitals for collecting the bacterial isolates.

Abbreviations

ARG Antibiotic resistance gene

ABHE A. baumannii isolated from hospital environment

CARD Comprehensive antibiotic resistance database

CRAB Carbapenem-resistant A. baumannii

MDRAB Multidrug-resistant A. baumannii

MLST Multilocus sequence typing

NDM New Delhi metallo-beta-lactamase

NRAB Non drug-resistant A. baumannii

SNP Single nucleotide polymorphism

VFDB Virulence factor database

XDRAB Extremely drug-resistant A. baumannii

WGS Whole-genome sequencing

Additional Information and Declarations

Competing Interests

Author Contributions

Ethics

DNA Deposition

Data Availability

The authors declare that they have no competing interests.

Thawatchai Kitti conceived and designed the experiments, performed the experiments, analyzed the data, prepared figures and/or tables, authored or reviewed drafts of the article, and approved the final draft.

Suphattra Manrueang performed the experiments, authored or reviewed drafts of the article, and approved the final draft.

Udomluk Leungtongkam performed the experiments, analyzed the data, authored or reviewed drafts of the article, and approved the final draft.

Supat Khongfak performed the experiments, analyzed the data, prepared figures and/or tables, authored or reviewed drafts of the article, and approved the final draft.

Rapee Thummeepak performed the experiments, analyzed the data, prepared figures and/or tables, authored or reviewed drafts of the article, and approved the final draft.

Surat Wannalerdsakun conceived and designed the experiments, authored or reviewed drafts of the article, and approved the final draft.

Thanyasiri Jindayok conceived and designed the experiments, authored or reviewed drafts of the article, and approved the final draft.

Sutthirat Sitthisak conceived and designed the experiments, analyzed the data, prepared figures and/or tables, authored or reviewed drafts of the article, and approved the final draft.

The following information was supplied relating to ethical approvals (i.e., approving body and any reference numbers):

The protocol was approved by the Naresuan University Institutional Biosafety Committee (NUIBC MI62-09-42).

The following information was supplied regarding the deposition of DNA sequences:

The nucleotide sequences of the eight A. baumannii isolates are available at NCBI’s Sequence Read Archive (SRA): PRJNA862456.

The following information was supplied regarding data availability:

The raw data are available in the Supplemental Files.

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
