# Peer review of "Genomic relatedness and dissemination of blaNDM-5 among Acinetobacter baumannii isolated from hospital environments and clinical specimens in Thailand"

_PeerJ, doi:10.7717/peerj.14831_

## Round 0.1 · original submission · Minor Revisions

Dear authors,

Three experts have addressed your manuscript and overall, they find your findings interesting. One of the reviewers has detailed issues that can be addressed and the other two experts have only minor comments. Please, check carefully the attached revision by reviewer 3 and address them in full. Please provide a rebuttal letter responding to all the concerns.

Best regards,
Bernardo

Reviewer 1 ·

Basic reporting

No comment

Experimental design

No comment

Validity of the findings

No comment

Additional comments

• In this study, The genomic relatedness and spread of blaNDM-5 among Acinetobacter baumannii isolated from hospital surroundings and clinical specimens were studied. The paper is well written and commendable as It is the first report of an NDM-5-producing A. baumannii being found in hospital surroundings and clinical samples in Thailand.
• Title: the title needs to paraphrase for more clarity by referring to the region and the country from which these strains were isolated.
Abstract:
• The abstract offers an accurate summary of the paper, and the language used in the abstract is easy to read and understand.
• Line 48-49: Please rephrase the sentence for more clarity:
blaOXA-23 , blaNDM, and blaOXA-58 as (80.2%), (78.30%), and (0.94%), respectively.
• Line 50-51: Please rephrase the sentence for more clarity:
MDRAB and extremely drug-resistant A. baumannii (XDRAB) were found in 81.97% and 55.74%,
respectively.
• Keywords: Where are the keywords???

Introduction:
• Introduction: The authors provide sufficient context on the subject.
• Line 69, 70: The complete name of the abbreviation should be added just before its first mention only. especially multidrug resistant A. baumannii (MDRAB) and extensively drug- resistant A. baumannii (XDRAB) change as: especially MDRAB and XDRAB
• Line 82: Please mention the full name when it comes at the first time: New Delhi metallo-beta-lactamase (NDM).
• Line 97-98: please re write the sentence for more clarity: Whole-genome sequencing studies comparing distinct clinical isolates and environments isolates, change as: Whole-genome sequencing studies comparing distinct clinical and environmental isolates.
Materials and methods:
• Line 124 ,127: Please mention the Catalog number.
• Line 178-179: Please explain what the acronym CARD, VFDB stands for, you can add a paragraph with all the abbreviations used in your text.
• The authors did not refer to the statistical program used in this study, in addition to the fact that the statistic is very simple. A statistical analysis was supposed to compare these isolates for better clarity
Results: The findings are presented clearly, and the results are reliable.
• Lines 198-200: Please rephrase the sentence for more clarity, change as: The isolates associated with patient contact from the ventilator, bedsheet, patient table, humidifiers, ventilation, curtain, and, IV stand were (77.9%), (66.7%), (66.7%), (55.6%), (53.3%), (33.3%), (33.3%), (33.3%), (33.3%), and at 13.3%, respectively.
• Lines 201-203: Please rephrase the sentence for more clarity, change as: The isolates involved staff contact and other environments belonging to air, keyboard, counter, medical lab coats dressing, trolley, stethoscope, chart, door handles, and telephone at (60.0%), (53.3%), (46.7%), (42.9%), (33.3%), (26.7%), (26.7%), (6.7%), and (6.7%), respectively.
• Line 222-224: Please rephrase the sentence for more clarity, change as: MDRAB and XDRAB were found in 93.40%, and 44.34%, respectively. For A. baumannii from clinical samples, CRAB, XDRAB, MDRAB, and NRAB were found in 81.97%, 81.97%, 55.74%, and 16.39%, respectively (Table S3).
• The authors mention the terminology of isolates throughout the article, while it is more accurate to call them strains after their confirmation at the molecular level. The tables and figures are clear and comprehensible, however, abbreviations in tables should be indicated in the bottom margins of the tables
Discussion:
• When citing references, please consider the chronology from oldest to newest, some examples of this in the lines 356-357.
Conclusions: are clear, the study offers sample data for the authors to draw.
Grammar: Need some revisions along the manuscript.

Annotated reviews are not available for download in order to protect the identity of reviewers who chose to remain anonymous.

·

Basic reporting

This is porimarily a descriptive but interesting work related to the prevalence of A. baumannii in a hospital setting and partial pehotypic and genomic characterization of some isolates.
It is well written and structured, figures and tables are adequate and references are enough to support the information given.

Experimental design

the experimental design is adequate.

Please provide a rational for the selection of the 8 isolates that were sequenced and a figure showing the comparison of those genomes and a reference strain such as ATCC17978.

Validity of the findings

Data shown is OK, but please provide a rational for the selection of the 8 isolates that were sequenced and a figure showing the comparison of those genomes and a reference strain such as ATCC17978.

Additional comments

minor concerns:

L. 74-75 "The major mechanisms of carbapenem resistance mechanisms" remove the second "mechanisms"

L 148 "were performed using primers (Table S1)." change to "were performed using the primers shown in Table S1"

L. 200 "13.3%" use "()"

L 215 "A. Baumannii " use "b"

L. 222 "baumanniifrom"

·

Basic reporting

Current work is interesting due to provide information about one of the most critical microorganisms associated with healthcare infections.
However has several details associated with language, in order to this is recommended the revision with a fluent English speaker,
Some comments have made directly in document

Experimental design

About methods is highly recommended that authors to check susceptibility breakpoints and the use of the most recent lietarute or guide such as CLSI 2022
Colistin method is no recommended by kirby bauer, only with broth microdilution to stimate MIC.
In sequencing and enssamble, annotation associated to several genes is no proper, e.g some of them are associated to macrolides however they belongs to aminoglycosides, ADC is annotated such as carbapenemase however it is cephalosporinase. All genes must be corroborated.

Validity of the findings

Document has several technical errors however all of them can be fixed and improved

---

## Round 0.2 · Minor Revisions

Dear authors,

I kindly request to attend the final comments made by reviewer 3 as soon as possible.

Warm regards

Reviewer 1 ·

Basic reporting

The current finding is intriguing since it sheds light on a major cause of healthcare-associated illnesses. The writers include enough background information, and the figures are appropriate, of great quality, and easy to understand.

Experimental design

The experimental design is adequate, especially after the authors responded to the comments raised by the reviewers related to this section and adhered to them.

Validity of the findings

The data is clear and sound. The conclusions are clear, the study offers sample data for the authors to draw.

Additional comments

The authors have addressed all the comments raised by the reviewers in the revised version of the manuscript. Therefore, this manuscript can be accepted for publication in the PeerJ.

·

Basic reporting

The authors performed and extensive improvement of the manuscript following the reviewer´s recommendations.

And addressed all my concerns.

Experimental design

The authors performed and extensive improvement of the manuscript following the reviewer´s recommendations.

And addressed all my concerns.

Validity of the findings

The authors performed and extensive improvement of the manuscript following the reviewer´s recommendations.

And addressed all my concerns.

Additional comments

The authors performed and extensive improvement of the manuscript following the reviewer´s recommendations.

And addressed all my concerns.

·

Basic reporting

Current work is interesting due to provide information about one of the most critical microorganisms associated with healthcare infections.
Authors have made alll changes and followed recomendations previously given.
Just some minimal comments
Page PDF 4. After Results -sic-A total of106 A. baumannii- of and 106 are together.
Line 143: Intermediately resistant. This definition it does not exist. Must be changed by Intermediate, the Term defined as resistant is when microorganisms is carrying any resistance mechanism. Intermediate talks about characteristics physic or chemical about hte molecule that cad support or not to reach the anatomical site where infection is.
Line 145. I´d like to see a brief definitiion of MDR, CR, and XDR.
Line 157, stil same doubt how representative is defined.
Line 181 were instead up was.
Line 196. Analysis instead up analyses.
Line 208-209, 212-213. TO change the way as numbers are presented. Comments in paper have been anoted directly in document.
Line 242. Technical pleonasm. -16S rRNA and rpoB genes were detected in all A. baumannii isolates- At least PCR with 16S primers it would has made in yeast, e..g, i expectec a negative reaction, however, all microorganism are bacteria then 16s must be present, same to rpob.
Line 243. Explain what happened with those 2 strains oxa51 negative.

Experimental design

Authors have made alll changes and followed recomendations previously given.

Validity of the findings

Authors have made alll changes and followed recomendations previously given.

Additional comments

Authors have made alll changes and followed recomendations previously given.

---

## Round 0.3 · accepted · Accept

Dear authors,
After the assessment of the expert reviewers, their consensus is that the manuscript has addressed all the issues they found and now is suitable for publication. I personally thank both the authors and the reviewers for the great job done. I wish you all the best for 2023.

With warm regards,
Bernardo

·

Basic reporting

Authors have follow all suggestion and made all pertinent changes

Experimental design

Authors have follow all suggestion and made all pertinent changes

Validity of the findings

Authors have follow all suggestion and made all pertinent changes

Additional comments

Authors have follow all suggestion and made all pertinent changes